# Charge State Effect of High Energy Ions on Material Modification in the Electronic Stopping Region

**Noriaki Matsunami [1,\*], Masao Sataka [2], Satoru Okayasu [2] and Bun Tsuchiya [1]**

1 Faculty of Science and Technology, Meijo University, Nagoya 468-8502, Japan; btsuchiya@meijo-u.ac.jp
2 Japan Atomic Energy Agency (JAEA), Tokai 319-1195, Japan; sataka@tac.tsukuba.ac.jp (M.S.); okayasu.satoru@jaea.go.jp (S.O.)
\* Correspondence: atsu20matsu@gmail.com

**Abstract:** It has been observed that modifications of non-metallic solids such as sputtering and surface morphology are induced by electronic excitation under high-energy ion impact and that these modifications depend on the charge of incident ions (charge-state effect or incident-charge effect). A simple model is described, consisting of an approximation to the mean-charge-evolution by saturation curves and the charge-dependent electronic stopping power, for the evaluation of the relative yield (e.g., electronic sputtering yield) of the non-equilibrium charge incidence over that of the equilibrium-charge incidence. It is found that the present model reasonably explains the charge state effect on the film thickness dependence of lattice disordering of $WO_3$. On the other hand, the model appears to be inadequate to explain the charge-state effect on the electronic sputtering of $WO_3$ and LiF. Brief descriptions are given for the charge-state effect on the electronic sputtering of $SiO_2$, $UO_2$ and $UF_4$, and surface morphology modification of poly-methyl-methacrylate (PMMA), mica and tetrahedral amorphous carbon (ta-C).

**Keywords:** charge-state effect; electronic excitation effect; high-energy ion; non-metallic solid; mean-charge evolution; sputtering; lattice disordering; surface morphology

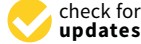



## 1. Introduction

Energetic ions lose their kinetic energies during passage through solid materials via collisions with electrons (inelastic collisions) and the nucleus (elastic collisions like billiard ball collisions) [1] and chapter 2 (Stopping Power of Ions in Matter) [2]. The elastic or nuclear collisions directly lead to the formation of the primary defect, i.e., Frenkel (interstitial and vacancy) pairs and usually, most of them are annealed out unless irradiation is performed at low temperature. On the other hand, the inelastic or electronic collisions (excitation of electrons and ionization) generally end up as de-excitation (with or without radiation emission) and heating of materials. However, for the ions with energies larger than ~0.1 MeV/u (the electronic stopping region), where the electronic stopping power (defined as the energy loss per unit path length via inelastic collisions) is dominant over the nuclear stopping power (the energy loss per unit path length due to elastic collisions), modifications of non-metallic solids induced by electronic energy deposition under the ion impact have been observed, for example, track formation in LiF (single crystal) [3], muscovite mica $(KAl_3Si_3O_{10}(OH)_2)$ [4], $SiO_2$ (crystalline quartz), glass (amorphous)-$V_2O_5$ (P-doped) and other insulating solids [5,6], and latent track (radius of several nm) in mica, $SiO_2$ (quartz), $Al_2O_3$ (crystalline sapphire), garnets [7], InP single crystal [8], amorphous-$Si_3N_4$ [9]. In addition, surface morphology modifications (formation of hillocks and craters) have been observed in poly-methyl-methacrylate (PMMA) [10], tetrahedral amorphous carbon [11] and mica [12]. Tracks and surface morphology have been observed by transmission electron microscopy (TEM) and atomic force microscopy (AFM). Moreover, the electronic sputtering (erosion of solid materials) caused by the electronic energy deposition has been observed for a variety of insulating and semiconducting compound solids: $UO_2$

(film) [13] and $UO_2$ (sintered disk) [14,15] by $^{235}U$ fission fragments induced by thermal neutron and sintered-$UO_2$ by ions [16–18], frozen films of $H_2O$ [19–22], Xe, $CO_2$ and $SF_6$ [21], Ar, $N_2$ and CO ices [23], $H_2O$ ice [24,25], $CO_2$ ice [26], $UF_4$ [18,27,28], $CaF_2$ [28], $SiO_2$ [29–35], $LiNbO_3$ [29], LiF (single crystal) [35,36], KBr (single crystal) [37], $Si_3N_4$ (amorphous film) [29,38], AlN [38], $Cu_3N$ [37,39], $Al_2O_3$ (single crystal) [29,40], oxides ($SrTiO_3$ (single crystal and polycrystalline film) and $SrCeO_3$ [31,40], $CeO_2$, MgO (single crystal), $TiO_2$ (single crystal) and ZnO (single crystal and polycrystalline film) [40], $Y_2O_3$ and $ZrO_2$ (Y-doped single crystal) [38], $Cu_2O$ [37,41], CuO [42], $WO_3$ [43], $Fe_2O_3$ [44] and SiC (single crystal) [37]. Here, $SrCeO_3$, $CeO_2$, $Y_2O_3$, $Cu_2O$, CuO, $WO_3$ and $Fe_2O_3$ are polycrystalline films. A typical method for sputtering measurements is that sputtered atoms are accumulated in a collector (or catcher) placed near the samples and followed by ion beam analysis of sputtered atoms in the collector.

The sputtering yields, $Y_{sp}$ (defined as number of ejected atoms from materials per incident ion) in the electronic stopping region are larger by $10–10^3$ than the sputtering yields due to the elastic collisions [19,27,29,31,33,37,38,40–44], which are calculated assuming the linear dependence on the nuclear stopping power ($S_n$) [45] and the yields $Y_{sp}$ do not scale with $S_n$. Furthermore, the sputtering of frozen Xe films [21] and water ice films [25] has been observed by low energy electron impact, contrary to the anticipation of no atomic displacement. These results confirm that sputtering or atomic displacement near the material surface is induced by electronic energy deposition (called electronic sputtering). It is mentioned that electronic-energy deposition effects on atomic displacement are indirect processes contrary to the direct elastic-collision effects. It is also noted that the electronic sputtering yields under the incidence of the equilibrium-charge ions follow the power-law of the electronic-stopping power ($S_e$): $Y_{sp} = (BS_e)^n$ with $1 \leq n \leq 4.6$, B being a material-dependent constant. This indicates that the electronic stopping power is appropriate for representing the electronic sputtering. Experimentally, the equilibrium-charge incidence is usually achieved by the insertion of thin foils such as carbon before ions hit samples. Stoichiometric sputtering has been observed for most of the cases. Deviation from the stoichiometric sputtering has been reported for $YBa_2Cu_3O_7$ [46], $Gd_3Ga_5O_{12}$ and $Y_3Fe_5O_{12}$ [47], and $CaF_2$, $LaF_3$ and $UF_4$ [28]. Moreover, sputtering yields do not seriously depend on the material phase and morphology (crystalline films, amorphous films, bulk single crystals, bulk polycrystals, etc.), e.g., sputtering yields of ZnO film and ZnO single crystals agree within 20% [40,48]. No appreciable difference has been observed for amorphous-$SiO_2$ (a-$SiO_2$), polycrystalline-$SiO_2$ films and single crystal of $SiO_2$ (c-$SiO_2$) [29,33], while it has been reported that the sputtering yields of c-$SiO_2$ are smaller by a factor of 3 than those of a-$SiO_2$ [34], and further study is required to resolve this issue.

The incident charge of ions is not always the same as the equilibrium-charge, intentionally or not. The mean-charge of ions (the average of the ion charge fractions) evolves during ions travel through materials and reaches the steady-state equilibrium-charge via electron loss and capture processes (Section 2). Thus, it is anticipated that material modification induced by the electronic energy deposition depends on the charge of incident ions, since the electronic stopping power depends on the ion charge (charge-state effect). Indeed, the charge-state effect has been observed on surface morphology modification [10–12], the electronic sputtering of $WO_3$ [43], LiF [36], $SiO_2$ [30,32], $UO_2$ [18] and $UF_4$ [27], and lattice disordering of $WO_3$ [49]. For example, when the incident charge is smaller than the equilibrium-charge, considerably lower sputtering yields have been observed [36,43]. However, the charge-state effect has not been well depicted. This paper concerns the charge-state effect of high-energy ion impact on material modification in the non-relativistic electronic-stopping region. Other effects (elastic collision effects, etc.) including some techniques in ion impact experiments are found in [50], effects of the potential-energy carried by low-velocity highly-charged ions on the sputtering [51,52], nano-structuring (including track formation, etc.) [53] and disordering of graphite structure in graphite and graphene by ion irradiation [54].

The object of the present paper is the quantitative modeling of the charge-state effects in the non-relativistic electronic-stopping region on the sputtering, lattice disordering and surface morphology modification. The charge-state effect can be qualitatively understood in terms of charge-dependent electronic-stopping power ($S_e$), which can be calculated using the CasP code for monatomic targets [55]. The accuracy of $S_e$ under the equilibrium-charge incidence is estimated to be 10%, e.g., Be through U ions in Ag solid target [56]. Two more physical quantities are required for a quantitative understanding of the charge-state effect: the equilibrium-charge and the mean-charge evolution of ions along the ion path. Basically, these could be evaluated from the charge-changing processes, if the electron loss and capture cross-sections are known (chapters 6 and 4 in [2]). Alternatively, the equilibrium-charge can be estimated using the tabulation of Wittkower et al. [57] and that of Shima et al. [58], and the empirical formulas of Ziegler et al. [1] and Schiwietz et al. [59]. The mean-charge evolution has been studied for S and C ions in carbon foil (Imai et al.) [60,61], and W ions in carbon foil in chapter 3 (Evolution of the Projectile Charge-State Fractions in Matter) [2]. However, the data of the mean-charge evolution are not available for compound solids concerned in this study. Taking these into account, a simple analytical model [49] is described in the next section. In the model, the mean-charge dependence on the depth is approximated by a saturation curve and Bragg's additive rule is applied to the electronic-stopping power for compound targets. The results and discussion follow in Section 3. A summary of discussions and conclusions are described in Sections 4 and 5.

## 2. Analytical Model

An analytical model [49] is reproduced to evaluate the charge-state effect on the electronic sputtering and XRD intensity modification. The model is based on the saturation approximation to the mean-charge ($Q_m$) evolution from the initial-charge, i.e., the incident charge ($Q_o$) to the steady-state equilibrium-charge ($Q_{eq}$), ($Q_m$, $Q_o$ and $Q_{eq} > 0$ in this study) and charge-dependent electronic-stopping power ($S_e$). There are a few reports on the mean-charge evolution in solids, 2 MeV/u S and C ions in carbon [60,61] and the former is considered here. Firstly, in Table 1, the experimental equilibrium-charge ($Q_{eq}$) is compared with the tabulation for carbon target by Shima et al. [58] and the empirical formula (3-38) by Ziegler et al. [1] where the contribution of the target electron velocity is safely discarded for high-energy ions (meaning that $Q_{eq}$ is independent of target),

$$Q_{eq}/Z_p = 1 - \exp(-0.95(y_r^{0.3} - 0.07)), \text{ with } y_r = V_p/V_o Z_p^{2/3}, \tag{1a}$$

and Equations (3) and (4) by Schiwietz et al. [59],

$$Q_{eq}/Z_p = (12y + y^4)/(0.07/y + 6 + 0.3y^{0.5} + 10.37y + y^4),$$

with

$$y = (v'Z_T^{-0.019v'}/1.68)^{1+1.8/Z_P} \text{ and } v' = (V_p/V_o)/Z_p^{0.52}. \tag{1b}$$

Here, $V_p$ is the velocity of projectile ions, $V_o$ is the Bohr velocity ($2.188 \times 10^8$ cm/s), $Z_p$ and $Z_T$ are the atomic number of incident ion and target atoms, respectively. The empirical values agree with the experimental value and the equilibrium charge in the gas phase [59] is also given for comparison with that in solids. An estimated accuracy of $Q_{eq}$ is a few % for solids [59]. $Q_{eq}$ values from Shima et al. [58] and those from (1b) reasonably agree with each other within several % for $WO_3$ (Sections 3.1 and 3.2) and for LiF (Section 3.3). Thus, the accuracy of $Q_{eq}$ is inferred to be several %.

**Table 1.** Equilibrium charge of S ions (2 MeV/u) in carbon foils. The value in the parenthesis in the last column is for gas phase.

| Experiment [a] | Empirical [b] | Empirical [c] | Empirical [d] |
|:---:|:---:|:---:|:---:|
| 12.68 | 12.63 | 11.51 | 12.58 (11.63) |

(a) Imai et al. [60], (b) Shima et al. [58], (c) Ziegler et al. [1], (d) Schiwietz et al. [59].

Secondly, the saturation approximation to the mean-charge ($Q_m$) evolution ($Q_m$ vs. depth X) is examined for 2 MeV/u $^{32}$S ions ($S^{+7}$ incidence) in carbon foils [60] and the calculated $Q_m$ by the Equation (2a) and (2b) is compared with the experimental result in Figure 1.

$$Q_m = Q_o + (Q_{eq} - Q_o) (1 - \exp(-X/L)) = Q_{eq} - (Q_{eq} - Q_o) \exp(-X/L), \quad (2a)$$

and

$$Q_m = Q_{eq} - \Delta Q_1 \exp(-X/L_1) - \Delta Q_2 \exp(-X/L_2) \text{ with } \Delta Q_1 + \Delta Q_2 = Q_{eq} - Q_o. \quad (2b)$$

Here, $Q_o$ is the incident charge and L is the characteristic length to attain the equilibrium charge. L of 7.5 nm is determined by fitting to Equation (2a) and it is related to the charge-changing cross-sections (electron-loss cross-section, in this case, see Table 2 for the relevant cross-sections) as discussed below. It is seen in Figure 1 that the single-saturation approximation (Equation (2a)) tolerably fits the experiment and the sum of two-saturation approximation (Equation (2b)) fits better. The electron-loss cross-section ($\sigma_L$) corresponding to L = 7.5 nm is obtained to be $0.13 \times 10^{-16}$ cm$^2$ ($\sigma_L = 1/LN$), where N is the C density of $10^{23}$ cm$^{-3}$ (2 g cm$^{-3}$). This value is smaller by a factor of 4 than the empirical total electron loss of $0.505 \times 10^{-16}$ cm$^2$ (Shevelko et al.) [62] (and chapter 6 [2]) and comparable with the single-electron loss of $0.11 \times 10^{-16}$ cm$^2$ (DuBois et al.), where the target dependence ($Z_T^{2/3}$) have been taken into account [63,64] (Table 2). These imply that loss of 4–5 electrons is involved in the total (multi)-electron loss process. Here, the 1st ionization potential (IP) is taken to be 328.75 eV for $S^7$ [65], and for the single-electron loss, the effective number of projectile electrons available for removal $N_{eff} = 7$ ($2s^2 2p^5$, ignoring $1s^2$), considering the Bohr's criterion that the electrons with smaller orbital velocity than the projectile ion velocity are removed [66] and chapter 1 [1]. The IP values (the difference of the corresponding total atomic energies) after Rodrigues et al. [67]) agree with those from [65]. Single-electron capture cross-section is estimated, using the scaling rule by Schlachter et al. [68] (Table 2). At incidence ($Q_o = 7$), the single-electron loss cross-section is larger by a factor of 4 than the capture cross-section and thus the estimation of L = 7.5 nm (or $\sigma_L = 0.13 \times 10^{-16}$ cm$^2$) described above is tolerable. An accuracy of L is inferred to be ~20% in single-saturation approximation from the difference of L = 7.5 and 8.7 nm (Table 2) and comparison of the single-saturation approximation with the experimental result in Figure 1. In the case of two-saturation approximation, we choose $L_1 = 2$ nm corresponding to the total loss cross-section $\sigma_L = 0.505 \times 10^{-16}$ cm$^2$, $L_2 = 17$ nm corresponding to the single-electron loss cross section $\sigma_L = 0.058 \times 10^{-16}$ cm$^2$ for $S^{10}$ (middle of $Q_o = 7$ and $Q_{eq} = 12.68$) with IP = 504.8 eV, $N_{eff} = 4$ ($2s^2 2p^2$) and $\Delta Q_1 = \Delta Q_2 = (Q_{eq} - Q_o)/2 = 2.84$. The single-electron capture cross-section of $0.072 \times 10^{-16}$ cm$^2$ (Q = 10) is comparable with the single-electron loss cross-section. Even though complications are involved in the mean-charge evolution, reasonable fit of the two-saturation curves shown in Figure 1 indicates that the electron loss process (multi-electron loss at shallow depth and one-electron loss at deeper region) reproduces the experimental mean-charge evolution. The accuracy of the two-saturation approximation is estimated to be ~10%, better than that of the single-saturation approximation. As mentioned in the introduction, the experimental data of the mean-charge evolution for compound solid targets concerned in this study are not usually available and we adopt the saturation approximation (Equation (2a) or (2b)) hereafter.

The relative yield (RY) (RY is defined as the yields under non-equilibrium charge incidence divided by those under the equilibrium-charge incidence) is calculated for

the sputtering yield, XRD degradation per unit ion fluence with the incident charge $Q_o$ smaller than the equilibrium charge $Q_{eq}$ using a simple model [49]. Here, the sputtering yields, etc., with the equilibrium-charge incidence follow the n-th power on the electronic stopping power $S_e$, which has been experimentally observed. $S_e(Q_{eq})$ is calculated by TRIM/SRIM code (based on the dielectric response to the projectile ion with the local density approximation for the electron density of target atoms and experimental data) [1] and CasP code (impact-parameter-dependent perturbation calculation as in the Bethe formula) [55]. With Equation (2a) or (2b), one finds the relative yield:

$$RY = (1/X) \int^x [S_e(Q_m(x)/S_{eq}]^n dX, \text{ with } S_{eq} = S_e(Q_{eq}). \tag{3a}$$

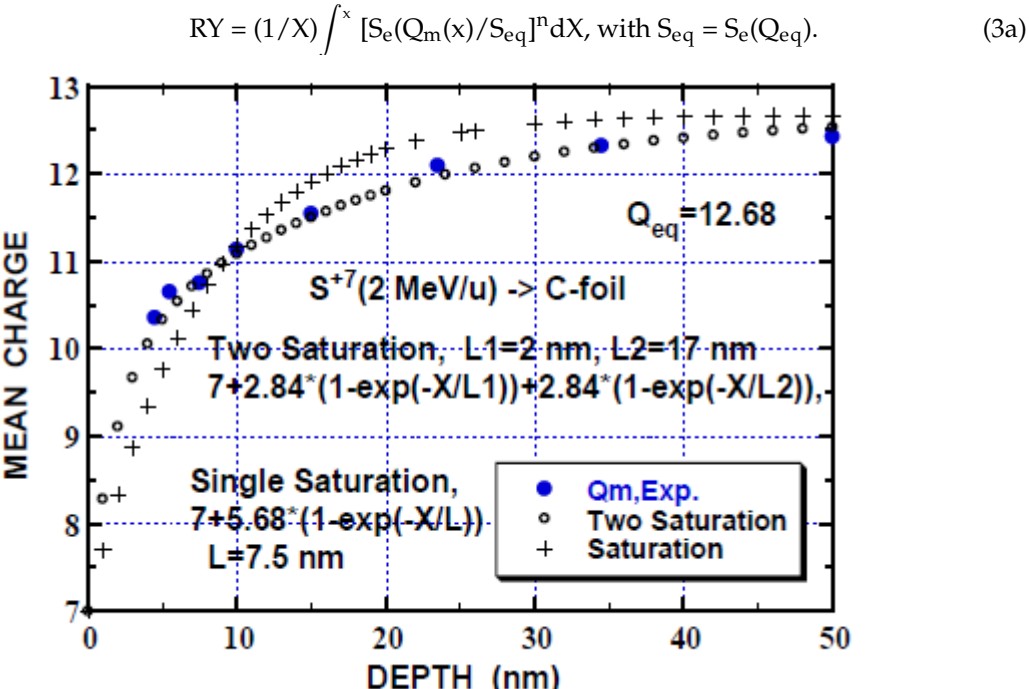

**Figure 1.** Mean-charge evolution of S (2 MeV/u) ions along the depth in carbon foils for the incident charge $Q_o$ =7 and equilibrium charge ($Q_{eq}$ =12.68) [60]: Sum of two saturation-curves approximation (Equation (2b)) (o) and single saturation-curve fit (Equation (2a)) (+) to the experimental mean-charge evolution (•). L is the characteristic length for attaining the equilibrium charge.

**Table 2.** Total electron loss cross section ($\sigma_{LT}$) after [62], single electron loss cross section ($\sigma_{L1}$) after [63,64] and single electron capture cross section from [68] in $10^{-16}$ cm$^2$ for 2 MeV/u $^{32}$S with charge state Q relevant to this study. IP is the first ionization potential for S$^{+Q}$ state [65] and $N_{eff}$ is the number of removable electrons. L is the characteristic length for attaining the equilibrium charge ($L_T = 1/\sigma_{LT}N$, $L_S = 1/\sigma_{L1}N$) corresponding to the total and single electron loss, N = $10^{23}$ cm$^{-3}$ (carbon density, 2 g cm$^{-3}$). The electronic configuration contributing to $N_{eff}$ for the S projectile ions at a given charge state is denoted in the parenthesis. The projectile velocity ($V_p$) divided by $V_o$ is 8.94 and the kinetic energy of the electron with $V_p$ is 1088 eV.

| Q | IP (eV) | $\sigma_{LT}$ $10^{-16}$cm$^2$ | $L_T$ (nm) | $N_{eff}$ | $\sigma_{L1}$ $10^{-16}$cm$^2$ | Ls(nm) | $\sigma_C$ $10^{-16}$cm$^2$ |
|---|---|---|---|---|---|---|---|
| 7 | 328.75 | 0.505 | 2.0 | 7($2s^22p^5$) | 0.114 | 8.7 | 0.029 |
| 10 | 504.8 | 0.344 | 2.9 | 4($2s^22p^2$) | 0.058 | 17.3 | 0.072 |
| 12 | 652.2 | 0.253 | 4.0 | 2($2s^2$) | 0.033 | 30 | 0.11 |
| 13 | 707 | 0.226 | 4.4 | 1($2s^1$) | 0.0227 | 44 | 0.13 |

Furthermore, when the power-law fit to the mean-charge ($Q_m$) dependence of the electronic stopping power $S_e$ is applicable such that $S_e$ is proportional to $Q_m{}^k$, Equation (3a) is rewritten as

$$RY = (1/X) \int^x [Q_m/Q_{eq}]^{nk} dX. \tag{3b}$$

At the limit of $X = 0$,

$$RY \, (X = 0) = (S_e(Q_o)/S_{eq})^n = (Q_o/Q_{eq})^{\,k}. \tag{3c}$$

As a summary of the model, using the equilibrium charge from the empirical value [58] or Equation (1b), saturation approximation to the mean-charge ($Q_m$) with the characteristic length L from the empirical formula of electron loss cross-sections [63,64] or capture cross-sections [68], experimentally obtained n value and k value for $S_e \propto Q_m{}^k$, RY is numerically evaluated (Equation (3a) or (3b)) and compared with the experimental results in Section 3.

## 3. Results and Discussion

Charge state effects are described as follows: the lattice disordering (XRD intensity degradation) of $WO_3$ films (Section 3.1), electronic sputtering of $WO_3$ films (Section 3.2), electronic sputtering of LiF (Section 3.3), electronic sputtering of $SiO_2$, $UO_2$ and $UF_4$ (Section 3.4). The charge-state effect on the surface morphology of PMMA, mica and ta-C is also discussed (Section 3.5).

### 3.1. Lattice Disordering of $WO_3$

The charge-state effect was observed on the degradation of XRD intensity (Cu-$K_\alpha$) for ultra-thin $WO_3$ polycrystalline films (a few nm to 30 nm) prepared by oxidation of W layers on MgO substrate in air at 520 °C [49]. Two strong diffraction peaks were observed at ~48° and ~23° depending on the film thickness [49,69], and the former diffraction is concerned in this paper. X-ray attenuation length for Cu-$K_\alpha$ of 8 keV is estimated to be 10 μm [70] and it does not play any role in such thin films. The crystal structure is orthorhombic or monoclinic and the films continuously or smoothly grew on the MgO substrate according to atomic force microscopy (AFM). The film thickness was obtained by Rutherford backscattering spectrometry of 1.8 MeV He. Figure 2a shows the XRD patterns of the diffraction angle of ~48° for unirradiated and irradiated $WO_3$ films by 90 MeV Ni ions ($0.48 \times 10^{12}$ cm$^{-2}$) without carbon foil (incident charge $Q_o = 10$) and with carbon foil of 100 nm (equilibrium-charge incidence), illustrating that the decrease of the XRD intensity depends on the incident charge. Normal incidence was employed. It is found that the degradation of the XRD peak intensity is proportional to the ion fluence, and then one obtains its slope, i.e., the XRD intensity decrease per unit ion fluence. The ratio of the XRD intensity degradation with the non-equilibrium charge incidence (Ni$^{10}$ and Xe$^{14}$) over that with the equilibrium-charge incidence is plotted as a function of the film thickness (Figure 2b). We examine whether the thickness dependence can be understood as the mean-charge evolution combined with the charge-dependent electronic stopping power or not. The characteristic length to attain the equilibrium charge ($Q_{eq}$) and the mean-charge evolution are evaluated using the empirical formulas of electron-loss cross-sections and the single-saturation approximation, respectively, as described in Section 2. Then, with the charge-dependent electronic stopping power (CasP) [55], the thickness dependence of the relative yields is calculated and compared with the experimental results.

In order to evaluate the film thickness dependence of the XRD intensity degradation (Figure 2b), i.e., one of the charge state effects, we utilize the experimental results of the XRD intensity degradation under the equilibrium-charge incidence as the function of the electronic stopping power ($S_e$) (Figure 3) as well as the equilibrium charge ($Q_{eq}$) and the mean-charge evolution. $Q_{eq}$ and $S_e$ are given in Table 3. Here, Bragg's additive rule is applied to obtain $S_e$. The contribution of oxygen to $S_e$ is 40–50%. $Q_{eq}{}^{(b)}$ is calculated such that $S_e(Q_{eq}{}^{(b)})$ of W equals $S_e$ of W with the equilibrium charge in the CasP code. Carbon foil (100 nm) is inserted to achieve the equilibrium charge and the energy loss in the carbon

foil is estimated to be 1, 1, 2 and less than 1 MeV for 90 MeV Ni, 100 MeV Xe, 200 MeV Xe and 60 MeV Ar ions [1]. The change in $S_e$ by insertion of the carbon foil is less than a few % and is negligibly small. $Q_{eq}^{(b)}$ is smaller than $Q_{eq}^{(c)}$ (CasP, Equation (1b)) and the latter is close to the value by Shima et al. [58]. $Q_{eq}^{(d)}$ by TRIM (Equation (1a)) is fairly smaller than the others. $S_e$ at $Q_{eq}$ by both CasP and TRIM is comparable and $S_e$ by TRIM 1997 agrees with that of SRIM 2013 (available on the web) within a few %, except for 90 MeV Ni ions, which differ by 10%. The exception for Ni does not seriously affect the following discussions. CasP estimation gives a fairly smaller $Q_{eq}$ for the gas phase than that for the solid phase. CasP 5.2 is employed throughout this paper.

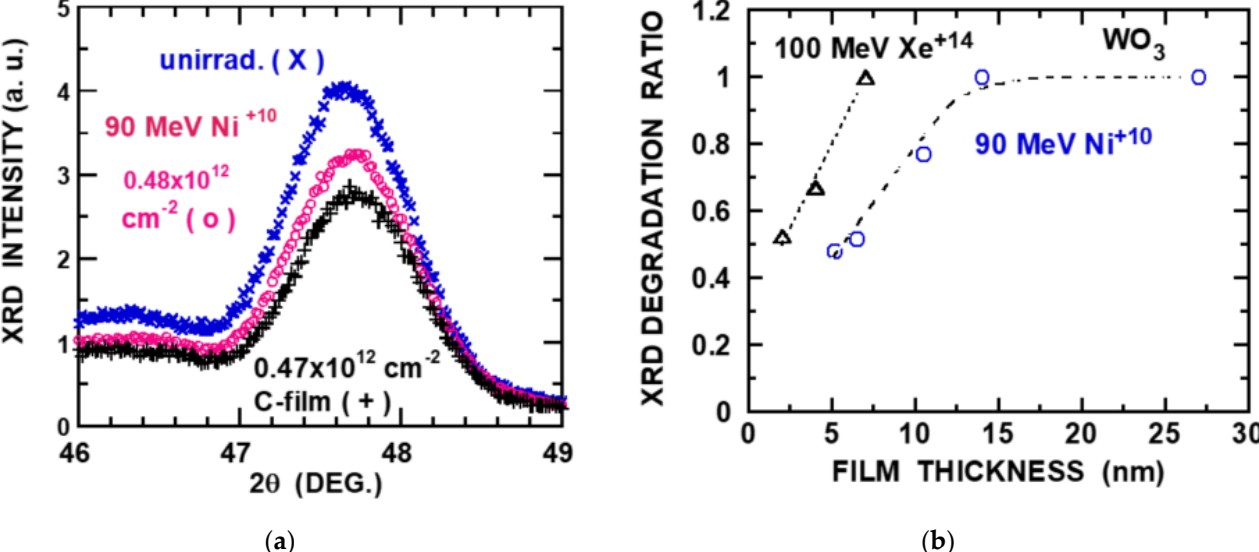

(a)                                      (b)

**Figure 2.** (a) XRD patterns of $WO_3$ films on MgO: unirradiated (x), 90 MeV $Ni^{+10}$ ions at $0.48 \times 10^{12}$ cm$^{-2}$ (o) and 90 MeV Ni ions through carbon foils (100 nm), i.e., equilibrium-charge incidence at $0.47 \times 10^{12}$ cm$^{-2}$ (+). Film thickness is ~6.5 nm. (b) Ratio of XRD intensity degradation by irradiation under 90 MeV $Ni^{+10}$ incidence over that 90 MeV Ni ions with the equilibrium charge (o) and ratio of XRD intensity degradation by irradiation under 100 MeV $Xe^{+14}$ incidence over that 100 MeV Xe ions with the equilibrium charge (∆). Dot and dashed lines are guides for eyes. Experimental XRD data are from [49].

XRD intensity degradation $Y_{XRD}$ per unit fluence vs. $S_e$ under the equilibrium-charge incidence is shown in Figure 3, together with the electronic sputtering $Y_{SP}$ vs. $S_e$. Power-law well fits the experimental yields Y. $S_e$ dependence of both $Y_{XRD}$ and $Y_{SP}$ is similar, indicating that the same mechanism operates for the lattice disordering and sputtering, even though small and large displacements are anticipated to be involved in lattice disordering and sputtering, respectively. Use of $S_e$ calculated by TRIM gives a slightly larger exponent of the power-law fit than that by CasP. Charge-dependent $S_e$ is calculated by CasP and power-law fits are shown in Figure 4. One observes that CasP reproduces the experimental charge-dependent $S_e$ ($Q_o$ =6 to 10) with an accuracy of ~10% for 2 MeV/u Ne ions in C by Blazevic et al. [71]. The exponent of the power-law fits (Table 4) is less than unity and much smaller than 2 anticipated from the unscreened Coulomb interaction.

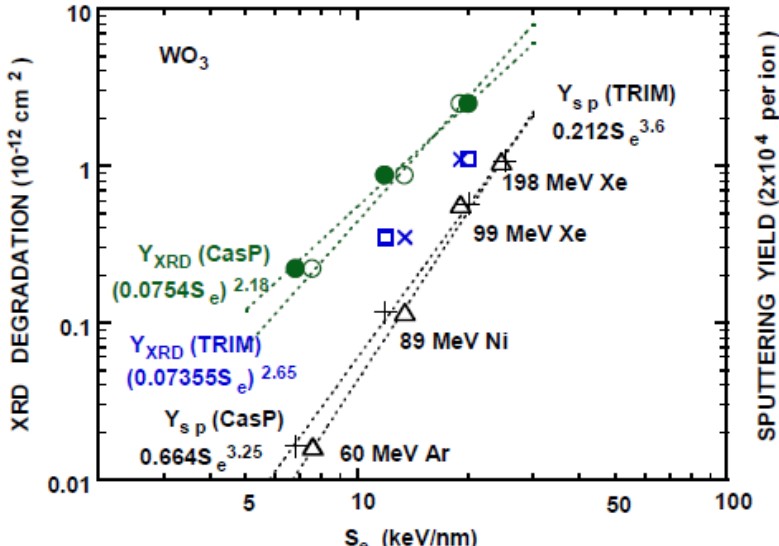

**Figure 3.** XRD intensity degradation of the diffraction angle of ~48° (o, •), 23° (x, □) and the electronic sputtering yields (Δ, +) by irradiation of ions with the equilibrium-charge incidence as the function of the electronic stopping power $S_e$, which are calculated by TRIM 1997 [1] (o, x, Δ) and CasP [55] (•, □, +). Power law fits to the XRD degradation (~48°) $Y_{XRD}$ and the electronic sputtering yield $Y_{SP}$ are indicated by the dot lines. From [49].

**Table 3.** Ion, energy (E in MeV), equilibrium charge ($Q_{eq}$), electronic stopping power ($S_e$ in keV/nm) for $WO_3$. W density in $WO_3$ is taken to be $1.87 \times 10^{22}$ cm$^{-3}$ (7.2 gcm$^{-3}$).

| Ion | E (MeV) | $Q_{eq}$ (a) | $Q_{eq}$ (b) | $Q_{eq}$ (c) | $Q_{eq}$ (d) | $S_e$ (e) | $S_e$ (f) (keV/nm) | $S_e$ (g) |
|---|---|---|---|---|---|---|---|---|
| $^{58}$Ni | 90 | 19 | 16.8 | 18.2(15.8) | 14.7 | 11.91 | 13.46 | 12.36 |
| $^{136}$Xe | 100 | 25 | 20.4 | 23.9(16.1) | 13.8 | 20.0 | 19.14 | 19.4 |
| $^{136}$Xe | 200 | 30 | 26 | 29.3(22.3) | 19.3 | 25.14 | 24.56 | 24.44 |
| $^{40}$Ar | 60 | 13 | 11.9 | 12.8(11.6) | 11.4 | 6.824 | 7.585 | 7.462 |

(a) Shima et al. [58], (b) $Q_{eq}$ is evaluated to match with the electronic stopping power of W using CasP [55] (see text), (c) Schiwietz et al. [59] (Equation (1b)) and the value in parenthesis is for gas phase [59], and $Q_{eq}$ is average of W and O values according to the composition, (d) Ziegler et al. [1] (Equation (1a)), (e) CasP [55], (f) TRIM 1997 [1], (g) SRIM 2013. CasP 5.2 is employed throughout this paper. Partly from [49].

The electron loss and capture cross-sections are estimated after [62–64] and [68], and given in Table 5a,b for 90 MeV Ni and 100 MeV Xe ions. Ionization potentials (IP) of Ni ions are from [65] and those of Xe are obtained to be the difference of the corresponding total atomic energies [67]. IPs from [65] agree well with those from [67] for Ni ions. $N_{eff}$ for the single electron loss cross-section [63] is evaluated considering Bohr's criterion and the electron orbital velocity or the kinetic energy (T) can be estimated from the binding energy (energy level in the bound state, BE or ionization potential) using the virial theorem (T = –BE for Coulomb potential and T > –BE for screened Coulomb potential). Careful estimation of the orbital velocity for use of Bohr's criterion would be desired.

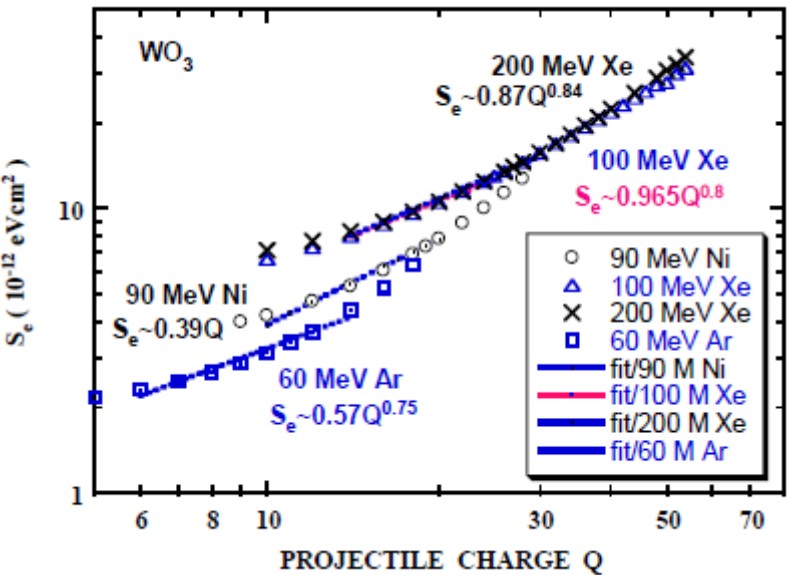

**Figure 4.** Electronic stopping power ($S_e$) as a function of the projectile ion charge Q for $WO_3$ calculated by CasP [55] using the Bragg additive rule for 200 MeV Xe, 100 MeV Xe, 90 MeV Ni and 60 MeV Ar. Power-law fits are also shown for Q = 14 to 30 (200 MeV Xe), Q = 14 to 25 (100 MeV Xe), Q = 10 to 20 (90 MeV Ni) and Q = 6 to 14 (60 MeV Ar).

**Table 4.** Ion, energy (E in MeV), exponent (k) of power-law fit that the charge(Q)-dependent electronic stopping power $S_e$ is proportional to $Q^k$ for $WO_3$. $S_e$ is calculated by CasP [55]. The region of the charge for the power-law fit is given in parenthesis.

| Ion | E (MeV) | k |
|-----|---------|---|
| $^{58}Ni$ | 90 | 1.0 (Q = 10–20) |
| $^{136}Xe$ | 100 | 0.8 (Q = 14–25) |
| $^{136}Xe$ | 200 | 0.84 (Q = 14–30) |
| $^{40}Ar$ | 60 | 0.75 (Q = 6–14) |

Now, the relative yield RY can be calculated using Equation (3b) and the results are shown in Figure 5. In the case of 90 MeV $Ni^{+10}$ ions, the total loss cross-section of $19 \times 10^{-16}$ $cm^2$ is larger by a factor of 17 than the single electron loss cross-section and hence, the total electron loss cross-section is unrealistic. One of the choices for L, n and k is that L = 4.8 nm corresponding to the single electron loss, and nk = 2.18 (n = 2.18 (Figure 3) and k = 1 (Table 4)). The calculated result of RY(X) reasonably reproduces the experimental thickness dependence of the XRD degradation, though the experiment shows stronger thickness dependence (Figure 5a). Another choice of L = 2.4 nm corresponding to multi-electron loss and nk = 3.0, e.g., n = 2.18 with k = 1.38 (stronger Q dependence of $S_e$ than CasP estimation) or n = 2.65 (TRIM result in Figure 3) with k = 1.13 gives slightly better agreement with the experiment. Reasonable agreement of the calculation with the experiment implies that estimation of the electron-loss cross-section [63,64] is sound. However, it is noted that the saturation curve for the charge evolution does not lead to the near-linear dependence of the experimental relative yield. This point will be discussed later. Similar results are seen in Figure 5b for 100 MeV Xe ions. In this case, L = 2.26 nm and 1.5 nm corresponding to two and three times of the single-electron loss cross-section (multi-electron loss), respectively, with nk = 1.774 (n = 2.18 and k = 0.8) are employed. In conclusion of this section, the experimental results of thickness dependence of the relative XRD degradation yield can be reasonably explained by the empirical cross-section of single or multi-electron loss (elucidated from presumably gas targets) with the

saturation approximation to the mean-charge evolution and power-law fit to the charge-dependent electronic stopping power. It is noted that the thickness dependence of the model calculation is weaker than that of the experiments.

**Table 5.** (a) Data of 90 MeV Ni$^Q$ for ion charge of Q in WO$_3$ relevant to this study. Total electron loss cross section ($\sigma_{LT}$) after [62], single electron loss cross section ($\sigma_{L1}$) after [63,64] and single electron capture ($\sigma_C$) [68] in $10^{-16}$ cm$^2$. IP is the first ionization potential for Ni$^Q$ state [65] and N$_{eff}$ is the number of removable electrons. $\sigma_{L1T} = \sigma_{L1}(W) + 3\sigma_{L1}(O)$ according to the WO$_3$ composition. L is the characteristic length for attaining the equilibrium-charge (L = $1/\sigma_{L1T}$N), N =$1.87 \times 10^{22}$ cm$^{-3}$ (W density in WO$_3$). The electronic configurations contributing to N$_{eff}$ for the Ni projectile ions are given in the parentheses. $V_p/V_o$ is 7.88 and the kinetic energy of the electron with $V_p$ is 844 eV. (b) Similar to (a) except for data of 100 MeV Xe$^Q$. IP for Xe$^Q$ is from [67]. The electronic configurations contributing to N$_{eff}$ for the Xe projectile ions are given in the parentheses. Vp/ Vo is 5.42 and the kinetic energy of the electron with Vp is 400 eV.

| Q | IP (eV) | $\sigma_{LT}$ W+O $10^{-16}$ cm$^2$ | N$_{eff}$ | $\sigma_{L1}$ W $10^{-16}$ cm$^2$ | $\sigma_{L1T}$ (W+O) | Ls (nm) | $\sigma_C$ $10^{-16}$ cm$^2$ |
|---|---|---|---|---|---|---|---|
| (a) 90 MeV $^{58}$Ni | | | | | | | |
| 10 | 321 | 18.7 | 8 ($3s^2 3p^5$) | 0.668 | 1.12 | 4.8 | 0.71 |
| 14 | 464 | 12.6 | 4 ($3s^2 3p^2$) | 0.337 | 0.567 | 9.4 | 1.34 |
| 17 | 607 | 8.73 | 1 ($3s^1$) | 0.142 | 0.239 | 22.4 | 1.94 |
| 18 | 1541 | 1.1 | 8 ($2s^2 2p^6$) | 0.10 | 0.17 | 31.5 | 2.17 |
| (b) 100 MeV $^{136}$Xe | | | | | | | |
| 14 | 343 | 20.0 | 12 ($4s^2 4p^6 4d^2 5s^2$) | 0.702 | 1.18 | 4.5 | 6.3 |
| 18 | 549 | 7.42 | 8 ($4s^2 4p^6$) | 0.332 | 0.557 | 9.6 | 10 |
| 20 | 616 | 5.5 | 6 ($4s^2 4p^4$) | 0.254 | 0.428 | 12.5 | 12 |
| 24 | 818 | 2.34 | 2 ($4s^2$) | 0.112 | 0.189 | 28 | 16 |

### 3.2. Electronic Sputtering of WO$_3$

The charge state effect has been also observed on the electronic sputtering yield by 90 MeV Ni ions, i.e., the yield under the non-equilibrium charge incidence (Ni$^{+10}$) is ~1/5 of that under the equilibrium-charge incidence [43]. Stoichiometric sputtering is observed for the equilibrium-charge incidence. As in the relative yield calculation of XRD degradation, the relative yield of the electronic sputtering yield is calculated with nk = 3.25 (n = 3.25 (Figure 3) and k = 1 (Table 4)) and the result is shown in Figure 6. The effective depth L'$_{SP}$ for the electronic sputtering is obtained to be 1.5 nm from the experimental yield of 1/5. This length is far smaller than the experimentally determined L$_{SP}$$^*$ = 40/2.3 = 17 nm (Figure 7) from [69]. Here, the factor of 2.3 is taken into account, since the length is 2.3 times the characteristic length (L) at RY = 0.9. It is noted that L$_{SP}$$^*$ is nearly independent of S$_e$ and characteristic of the material. The calculation with the extreme conditions of nk = 7.2 (n = 3.6 (TRIM result) in Figure 3 and k = 2) is also shown and this gives L'$_{SP}$ = 7 nm in improvement but poor agreement with the experimental result of L$_{SP}$$^*$ of 17 nm. Allowance of a factor of two for accuracy of the experimental yield and the calculation with the extreme conditions lead to L'$_{SP}$ = 13 nm, which is close to L$_{SP}$$^*$.

The experimental results that the electronic sputtering yields Y$_{SP}$ scale well with the electronic stopping power S$_e$ (Y$_{SP}$~S$_e$$^n$, as mentioned in the introduction) do not readily indicate the existence of the threshold S$_e$$^{SpTh}$, contrary to discussion [35]. However, this does not exclude the existence of S$_e$$^{SpTh}$, because of experimental difficulties, i.e., very low sputtering yield near S$_e$$^{SpTh}$. For WO$_3$, S$_e$$^{SpTh}$ is estimated to be below 6 keV/nm from Figure 3, though its existence should be carefully examined. At this stage, the simple model calculation is not adequate to explain the charge state effect on the electronic sputtering. A mechanism would be required that suppression of the electronic excitation

effect including the threshold and enhancement overcoming the saturation behavior of the mean-charge evolution. In other words, these imply that the single-electron loss cross-section (or the inverse of the length L) would be reduced in the near-surface region and enhanced in the deeper region. This mechanism would be also effective for the explanation of XRD degradation (nearly linear dependence on the film thickness). The $WO_3$ results are compared with LiF sputtering in the next section.

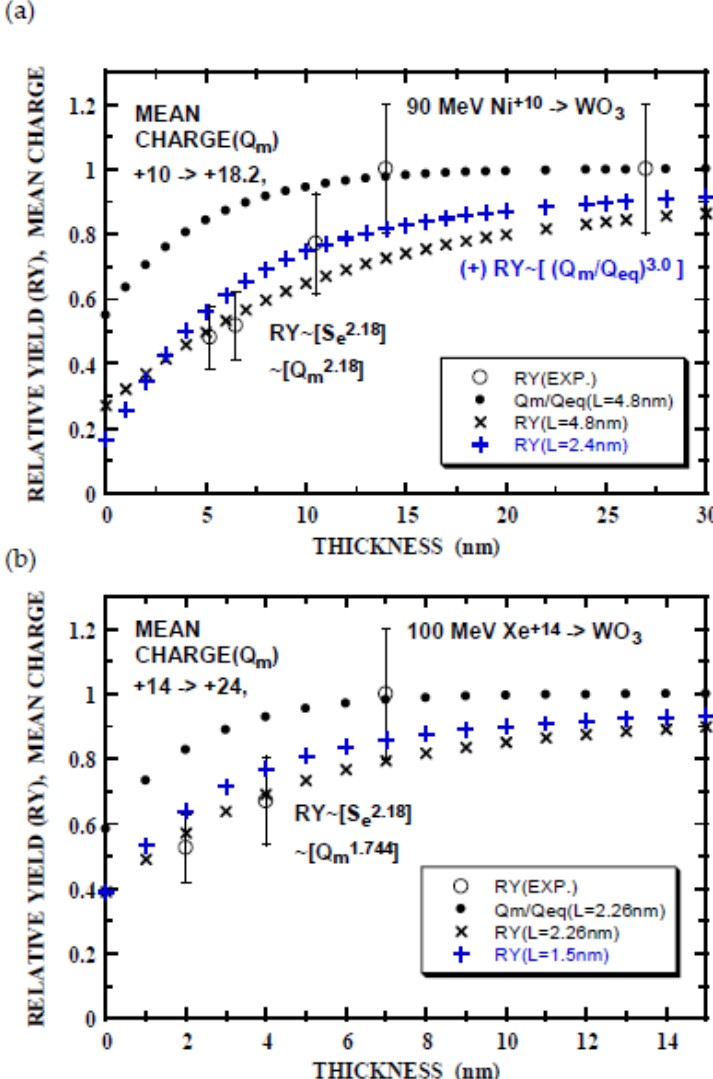

**Figure 5.** Thickness dependence of relative yield of XRD degradation of $WO_3$ films irradiated by 90 MeV $Ni^{+10}$ (**a**) and 100 MeV $Xe^{+14}$ ions (**b**). Open circles with an error of 20% are the experimental yields from Figure 2b. Small closed circles indicate the mean-charge evolution calculated by Equation (2a), for 90 MeV Ni, L = 4.8 nm, $Q_o$ = 10 and $Q_{eq}$ = 18.2, and for 100 MeV Xe, L = 2.26 nm, $Q_o$ = 14 and $Q_{eq}$ = 24. Relative yield RY is calculated using Equation (3b) (integrand is indicated by []). The parameters of the calculation for 90 MeV Ni ion indicated by x and + are as follows: (x) L = 4.8 nm, n = 2.18 and k = 1.0, and (+) L = 2.4 nm, nk = 3.0 (e.g., n = 2.18 and k = 1.38), respectively. Those for 100 MeV Xe ion are: (x) L = 2.26 nm with nk = 1.744 (n = 2.18 and k = 0.8), and (+) L = 1.5 nm with nk = 1.744 (n = 2.18 and k = 0.8), respectively.

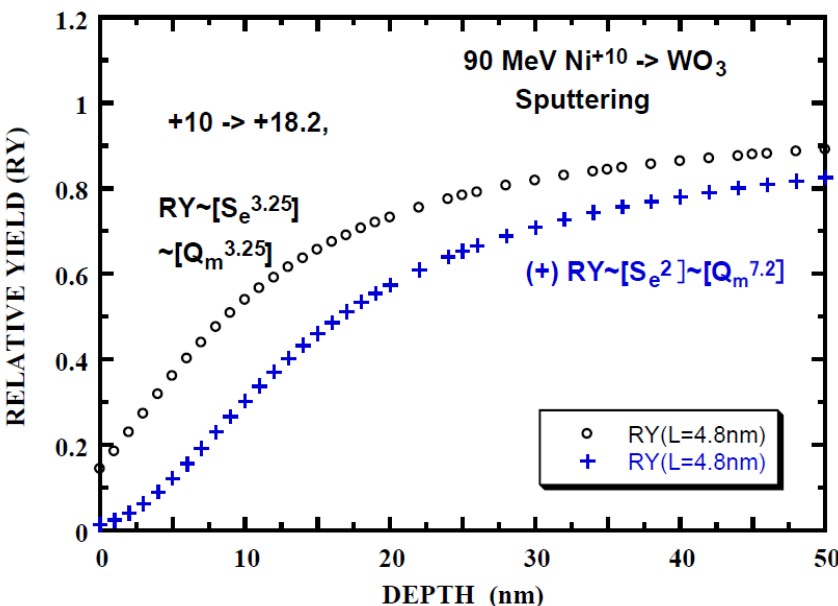

**Figure 6.** Depth dependence of relative yield (RY) of electronic sputtering of WO$_3$ films irradiated by 90 MeV Ni$^{+10}$. Open circles (o) and crosses (+) indicate RY calculated with nk = 3.25 (n = 3.25, k = 1) and nk = 7.2 (n = 3.6, k = 2), respectively, (integrand of Equation (3b) is indicated by [ ]). Mean-charge evolution calculated by Equation (2a) with L = 4.8 nm, Q$_o$ = 10 and Q$_{eq}$ = 18.2.

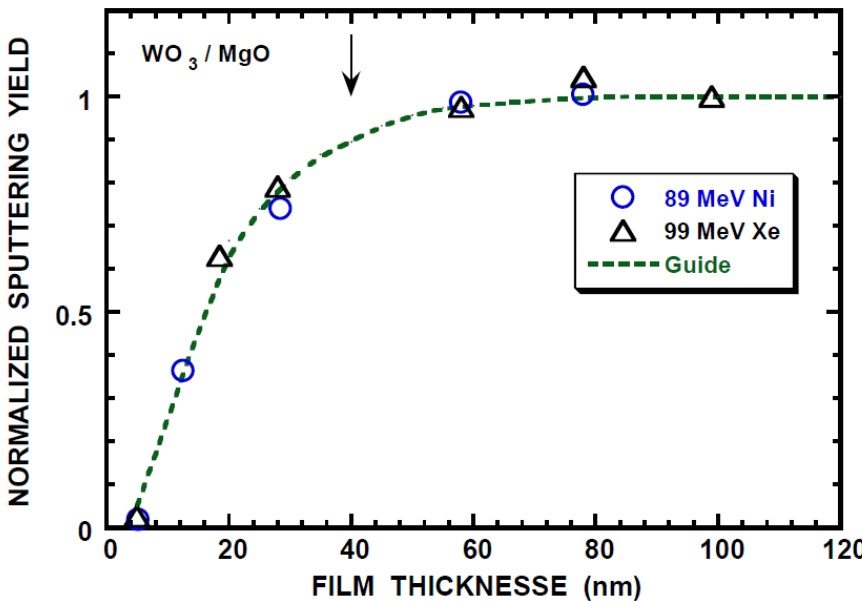

**Figure 7.** Sputtering yields of WO$_3$ by 89 MeV Ni (o) and 99 MeV Xe ($\Delta$) ions with the equilibrium-charge incidence vs. film thickness. The experimental yields after [69] are normalized to that at saturation (4.7 × 10$^3$ and 24 × 10$^3$ for 89 MeV Ni and 99 MeV Xe ions). The vertical line with arrow indicates the effective depth (40 nm) for the electronic sputtering and dashed line is a guide for eyes.

*3.3. Electronic Sputtering of LiF*

For the electronic sputtering of LiF single crystal, Toulemonde et al. [35] and Assmann et al. [36] have observed the strong peak around the exit angle of surface normal (anisotropic or jet-like component) with the broad isotropic component in the exit angle distribution of sputtered atoms, independent of the incident angle of ions. Sputtering appears to be stoichiometric within 10%. Furthermore, the charge state effect has been

reported on the electronic sputtering of LiF [36], as summarized in Table 6. Interestingly, the relative sputtering yield RY (the yield with non-equilibrium charge incidence divided by that with the equilibrium-charge incidence) depends on the incident angle ($\theta_1$) measured from the surface normal, i.e., RY increases with $\theta_1$. This implies that atoms escape easier in the near-surface regions than deeper regions. A similar situation is discussed for $WO_3$ sputtering (Section 3.2). $S_e$ of both TRIM 1997 and SRIM 2013 agree with each other, and $S_e$ of CasP reasonably agrees with others. It is mentioned again that Bragg's additive rule is applied to obtain the stopping power. $S_e$ of LiF by CasP (5.2 version) is found to be larger by a factor of roughly 30% than that given in Table 6 from [36]. This is considered to be the bonding effect or Bragg's deviation of –(5–30%) as reported for 0.7 MeV protons in $GdF_3$ and $HoF_3$ (Miranda et al.) [72], while there have been reports of –6% for 0.5 MeV/u He in LiF (Biersack et al.) [73] and ~10% larger $S_e$ compared to that of SRIM 2013 without Bragg's correction for 1 MeV p in LiF (Damache et al.) [74]. Absolute values of Q-dependent $S_e$ do not play a role in the present calculation of the relative yield.

**Table 6.** Data of LiF sputtering. Ion ($^{197}Au$, $^{132}I$ and $^{208}Pb$), energy (E in MeV), incident angle ($\theta_1 °$) measured from the surface normal, incident charge ($Q_o$), equilibrium–charge ($Q_{eq}$), equilibration length (Lps, reaching 90% of the electronic stopping power at the equilibrium–charge), ratios (RY) of the sputtering yield under the non-equilibrium-charge incidence over that the equilibrium-charge incidence for isotropic ($RY_{ISO}$), anisotropic ($RY_{ANI}$) components and total ($RY_{TO}$), electronic stopping power ($S_e$ in keV/nm) with the charge of $Q_o$ ($S_{eo} = S_e(Q_o)$) and $Q_{eq}$ ($S_{eq} = S_e(Q_{eq})$), from [36]. $S_{eq}$ by TRIM1997 (a) and SRIM 2013 (b) are given for comparison. The length (Lp, underlined) such that the experimental RY equals to the calculated RY is given below RY. Li density in LiF is taken to be $6.13 \times 10^{22}$ cm$^{-3}$ (2.64 gcm$^{-3}$).

| Ion | E (MeV) | $\theta_1$ | $Q_o$, $Q_{eq}$ | Lps (nm) | $RY_{ISO}$ | $RY_{ANI}$ Lp(nm) | $RY_{TO}$ | $S_{eo}$, $S_{eq}$ | $S_{eq}$ (a) | $S_{eq}$ (b) |
|---|---|---|---|---|---|---|---|---|---|---|
| $^{197}Au$ | 200 | 20 | 15, 31 | 18 | 0.0815 4.3 | 0.923 (176) | 0.226 11.2 | 9.7, 20 | 21.2 | 22.7 |
| $^{197}Au$ | 200 | 60 | 15, 31 | | 0.44 21.2 | 1.0 | 0.608 33.2 | | | |
| $^{132}I$ | 150 | 70 | 12, 25 | 19 | 0.44 *18.9* | 1.06 | 0.631 31.2 | 7.3, 15 | 16.0 | 16.8 |
| $^{208}Pb$ | 735 | 60 | 39, 51 | 170 | 0.41 73.1 | 0.64 174 | 0.507 109 | 20, 29 | 27.8 | 27.4 |
| $^{208}Pb$ | 735 | 60 | 47, 51 | 180 | 0.695 41.3 | 1.17 | 0.894 290 | 26, 29 | | |
| $^{208}Pb$ | 735 | 60 | 55, 51 | 200 | 1.715 8.0 | 1.666 16.4 | 1.69 12.2 | 33, 29 | | |
| $^{208}Pb$ | 936 | 60 | 23, 56 | 300 | 0.0181 21 | 0.0312 37.2 | 0.024 29 | 10, 30 | 28.1 | 27.0 |

Assmann et al. argued the path length contributing to the sputtering (Lp). Relative yields of anisotropic components ($RY_{ANI}$) is close to unity, meaning that Lp is comparable with Lps (path length that the electronic stopping power ($S_e$) attains 90% of $S_{eq}$) of 18 nm for 200 MeV Au[15], 150 MeV I[12] and ~200 nm for 735 MeV Pb ions. The situation is different in that $RY_{ANI}$ is smaller and larger than unity for 735 MeV Pb[39] and 735 MeV Pb[47], respectively, and that $RY_{ANI}$ is much smaller than unity for 936 MeV Pb[23]. Hence, the above argument does not generally hold. This point will be discussed after the calculation of RY. The relative yields of the isotropic component ($RY_{ISO}$) and total yields ($RY_{TO}$) are smaller than unity for the ions with smaller $Q_o$ than $Q_{eq}$ and these are larger than unity for 735 MeV Pb[55] ($Q_o > Q_{eq}$), indicating that Lp for these ions is smaller than Lps.

The equilibrium-charge [36] is compared with the empirical values in Table 7. For 200 MeV Au and 150 MeV I, $Q_{eq}$ in [36] is a little bit smaller than that estimated from Equation (1b) and tabulation [58]. The mean-charge evolution has been obtained by solving the rate equation with no description of the electron loss and capture cross-sections [36]. It

appears that the single-saturation or two-saturation approximation (Equation (2a) or (2b)) reasonably fits to the mean-charge evolution [36], as seen in Figures 8–10 for 200 MeV Au[15], 735 MeV Pb[55] and 936 MeV Pb[23] ions. The characteristic length L and the corresponding electron loss cross-section for $Q_o < Q_{eq}$ (or capture cross-section for $Q_o > Q_{eq}$) are given in Table 7. These cross-sections are compared with those from the empirical formulas of the electron loss and capture cross-sections [62–64,68] in Table 8. Ionization potentials are from [67]. For 200 MeV Au[15], 150 MeV I[12], 735 MeV Pb[39] and 735 MeV Pb[47] ion incidence, where the mean-charge increases from $Q_o$ along the path, the electron loss cross-sections ($\sigma_L$) corresponding to the characteristic length L (Table 7) accord with the empirical single-electron loss cross-sections indicated by values in Table 8 within 20%, except for the following. In the case of 936 MeV Pb[23] ion incidence, the empirical single-electron loss overestimates by more than a factor of 3 and the empirical total-electron loss cross sections are much larger than $\sigma_L$. The empirical single-electron capture cross-section ($1.48 \times 10^{-16}$ cm$^2$) is much larger (a factor of 60) than $\sigma_C$ of $0.0204 \times 10^{-16}$ cm$^2$ for 735 MeV Pb[55] ion incidence, where the mean-charge decreases from $Q_o$ along the ion path. One also observes that Lps (Table 6) = 2.4*L (Table 7) within 20%. The factor of 2.4 agrees with the factor of 2.3 at RY = 0.9, as described below.

**Table 7.** Ion, energy (E in MeV), equilibrium charge ($Q_{eq}$), parameters of the power-law fit to the charge (Q)-dependent electronic stopping power $S_e$ (keV/nm) and the range of Q in parenthesis for LiF. L is the characteristic length in the saturation fit (Equation (2a) or (2b)) to the mean-charge evolution and $\sigma$ is the corresponding electron loss and capture cross sections ($\sigma = 1/NL$) for increasing charge, except for decreasing charge ($Q_o = 55$–50.5, 735 MeV Pb denoted by *). N = $6.13 \times 10^{22}$ cm$^{-3}$ (Li density, see Table 6).

| Ion | E (MeV) | $Q_{eq}$ [(a),(b),(c)] | $S_e(Q) = BQ^K$ B, K (Q range) | L(Q Range) (nm) | $\sigma$ $10^{-16}$cm$^2$ |
|-----|---------|-------------------|-------------------------------|-----------------|------------------------|
| Au | 200 | 31, 34.7, 33 | 1.457, 0.7 (15–19.1) 0.3895, 1.147 (19.1–31) | 8 (15–31) | 0.204 |
| I | 150 | 25, 27.2, 28 | 1.33, 0.6855 (12–15.6) 0.3763, 1.145 (15.6–25) | 7 (12–15) | 0.233 |
| Pb | 735 | 51, 50.5, 50.5 | 0.098, 1.45 (39–55) | 60 (39–50.5) 75 (47–50.5) 70 (55–50.5) | 0.0272 0.218 0.0233 * |
| Pb | 936 | 56, 53.5, 54.5 | 0.436, 1.0 (23–36.29) 0.07918, 1.475 (36.29–56) | 35 & 150 (23–56) | 0.0204 |

(a) Assmann et al. [36], (b) Schiwietz et al. [59] (Equation (1b) for solid target) and (c) Shima et al. [58].

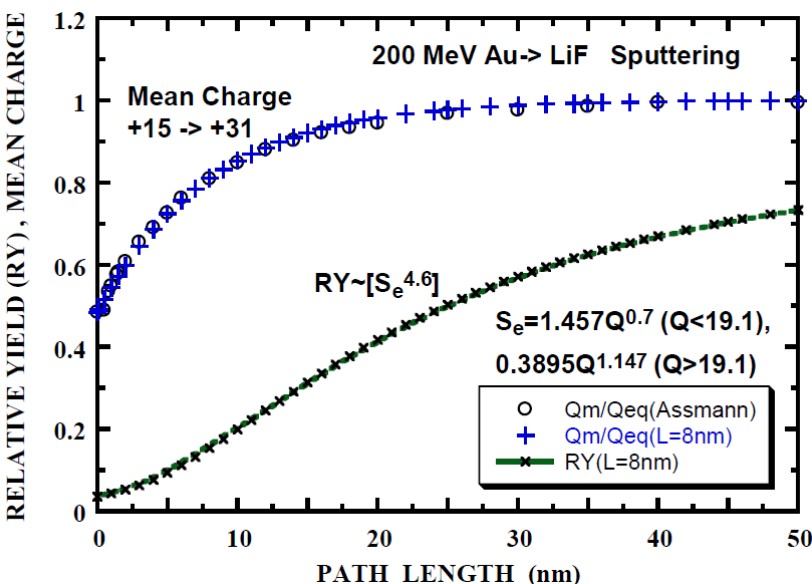

**Figure 8.** Relative yield (RY) of the electronic sputtering of LiF by 200 MeV Au$^{+15}$ (x) calculated with n = 4.6, $S_e$ = 1.457Q$^{0.7}$ (Q < 19.1) and $S_e$ = 0.3895Q$^{1.147}$ (Q > 19.1) as a function of path length. Open circles (o) is the mean-charge evolution from [36] normalized to the equilibrium-charge $Q_{eq}$ = 31 and saturation approximation (Equation (2a)) with L = 8 nm and $Q_o$ = 15 (+). Integrand for RY is indicated by [ ].

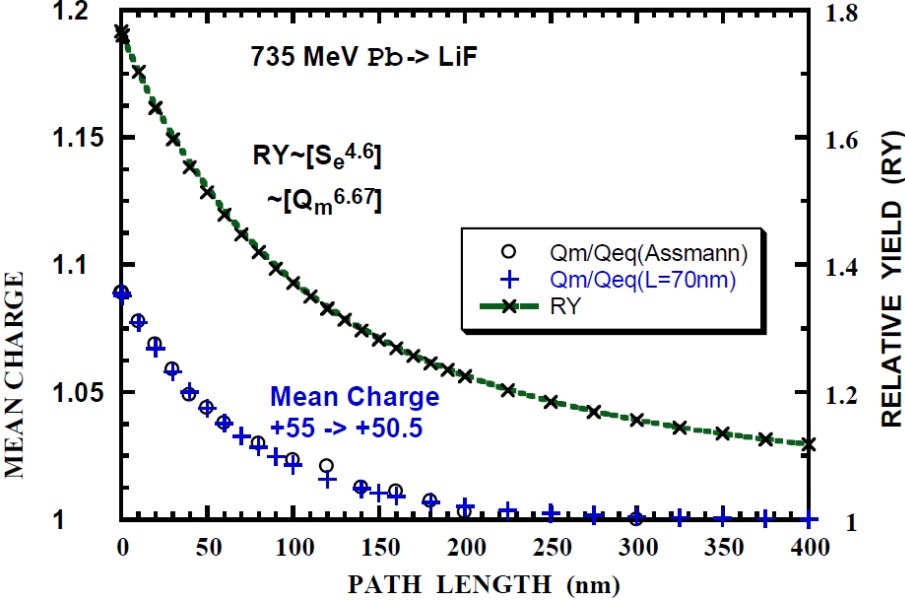

**Figure 9.** Relative yield (RY) of the electronic sputtering of LiF by 735 MeV Pb$^{+55}$ (x) calculated with n = 4.6, $S_e$ = 0.098Q$^{1.45}$ as a function of path length. Open circles (o) is the mean-charge evolution from [36] normalized to the equilibrium-charge $Q_{eq}$ = 50.5 and saturation approximation (+) (Equation (2a)) with L = 8 nm and $Q_o$ = 55. Integrand for RY is indicated by [ ].

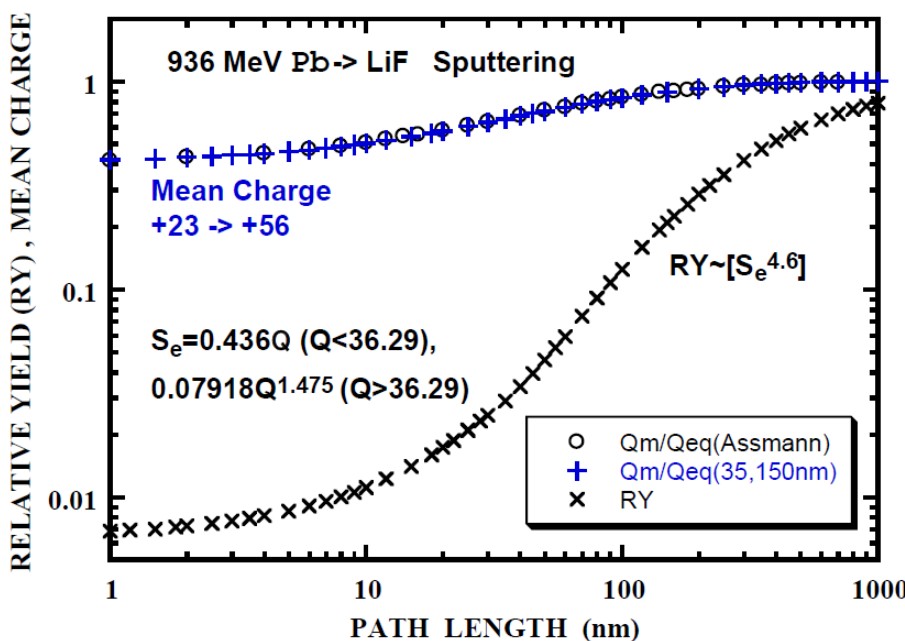

**Figure 10.** Relative yield (RY) of the electronic sputtering of LiF by 936 MeV $Pb^{+23}$ (x) calculated with n = 4.6, $S_e$ = 0.436Q (Q < 36.3) and $S_e$ = $0.07918Q^{1.475}$ (Q > 36.3) as a function of the path length. Open circles (o) is the mean-charge evolution from [36] normalized to the equilibrium-charge, $Q_{eq}$ = 56 and (+) Equation (2b), $Q_m/Q_{eq}$ = ($Q_{eq}$ − 17exp(-PL/35) − 16exp(-PL/150))/$Q_{eq}$, PL being the path length. Integrand for RY is indicated by [ ].

The relative sputtering yield (RY) is calculated using Equation (3a) or (3b) with the experimentally obtained value of n = 4.6 [36] and parameters of power-law fit to $S_e$(Q) given in Table 7. Typical results are shown in Figures 8–10. It is noted that the calculated RY does not exceed unity (Equations (3a) and (3b)). Hereafter Lp is defined as the path length such that the calculated RY is equals the experimental RY. Lp and RY are summarized in Table 9. Lp of 176 nm ($RY_{ANI}$ = 0.923, 200 MeV $Au^{15}$ with$\theta_1$ = 20° incidence) indicates that the calculated RY is applicable for RY < 0.9. Firstly, it appears that Lp corresponding to $RY_{ISO}$ is smaller by a factor of 1.5 than Lp ($RY_{TO}$), except for 200 MeV $Au^{15}$ ion with $\theta_1$ = 20° incidence and 735 MeV $Pb^{47}$ ion with $\theta_1$ = 60° incidence (RY is close to 0.9). Hereafter, $RY_{TO}$ is concerned. Secondly, it is found that Lp = 11 nm ($RY_{TO}$ = 0.226 for 200 MeV $Au^{15}$ ion with $\theta_1$ = 20° incidence, nearly normal incidence) is comparable with Lps/2.3 = 7.8 nm, considering the factor of 2.3 that Lps = 2.3L at RY = 0.9 (nearly corresponding to 90% $S_e$ of that at the equilibrium-charge) as described in the first paragraph, Section 3.2. Lp = 33 nm ($RY_{TO}$ = 0.608 for 200 MeV $Au^{15}$ ion with $\theta_1$ = 60°), Lp = 31 nm ($RY_{TO}$ = 0.631 for 150 MeV $I^{12}$ with $\theta_1$ = 70 °), Lp = 109 nm ($RY_{TO}$ = 0.507 for 735 MeV $Pb^{39}$ with $\theta_1$ = 60°) and Lp = 290 nm ($RY_{TO}$ = 0.894 for 735 MeV $Pb^{47}$ with$\theta_1$ = 60°) are larger by a factor of 1.5–4.3 than Lps/2.3. Lp = 12 nm ($RY_{TO}$ = 1.69 for 735 MeV $Pb^{55}$ with $\theta_1$ = 60) and Lp = 29 nm ($RY_{TO}$ = 0.031 for 936 MeV $Pb^{23}$ with $\theta_1$ = 60°) are smaller by a factor of 7–4.5 than Lps/2.3. For $Q_o$ < $Q_{eq}$ (the electron loss process is dominant), the characteristic depth (Lp·cos($\theta_1$)) corresponding to Lp($RY_{TO}$) is obtained to be 13.6 ± 3 nm for 4 cases among 6 (200 MeV Au, 150 MeV I and 936 MeV Pb). This implies the existence of the characteristic depth for the electronic sputtering independent of ion energy and species as in the case of $WO_3$ [69]. For 735 MeV $Pb^{55}$ ($Q_o$ > $Q_{eq}$, the electron capture process being dominant), the characteristic depth of 6 nm is smaller by a factor of 2 and the difference from the above-mentioned value of 13.6 nm could be partly due to the inaccuracy of the electron capture cross-section. For 735 MeV $Pb^{47}$ ($Q_o$ < $Q_{eq}$), the applicability is beyond the present model as already described. For 735 MeV $Pb^{47}$ ($Q_o$ < $Q_{eq}$), the characteristic depth of 54.5 nm is larger by a factor of 4 and the discrepancy could be partly due to the

inaccuracy of the electron loss cross-section. The thickness dependence of the sputtering yield for the determination of the effective path length or depth would be fruitful.

**Table 8.** Charge state Q in LiF relevant to this study, the first ionization potential (IP eV) [67], total electron loss cross section ($\sigma_{LT}$) [62], single electron loss cross section ($\sigma_{L1}$) [63,64] and single electron capture [68] in $10^{-16}$ cm$^2$ for 200 MeV Au, 150 MeV I, 735 MeV Pb and 936 MeV Pb with Q. $N_{eff}$ is the number of removable electrons. The electronic configurations contributing to $N_{eff}$ for the projectile ions are given in the parenthesis. $\sigma_{L1T} = \sigma_{L1}(Li) + \sigma_{L1}(F)$. Projectile velocity ($V_p$) divided by Bohr velocity ($V_o$) and the kinetic energy ($E_{KE}$) of the electron with $V_p$ are given in the parenthesis after the energy and ion.

| Q | IP (eV) | $\sigma_{LT}$ Li+F $10^{-16}$cm$^2$ | $N_{eff}$ | $\sigma_{L1}$ F $10^{-16}$cm$^2$ | $\sigma_{L1T}$ Li+F $10^{-16}$cm$^2$ | $\sigma_{C}$ Li+F $10^{-16}$cm$^2$ |
|---|---|---|---|---|---|---|
| 200 MeV $^{197}$Au ($V_p/V_o$ = 6.37, $E_{KE}$ = 552 eV) | | | | | | |
| 15 | 452 | 0.919 | 10 ($4f^{10}$) | 0.12 | 0.177 | 1.70 |
| 31 | 945 | 0.173 | 12 ($4d^{10}5s^2$) | 0.050 | 0.0747 | 6.82 |
| 150 MeV $^{132}$I ($V_p/V_o$ = 6.74, $E_{KE}$ = 618 eV) | | | | | | |
| 12 | 279 | 1.40 | 5 ($4d^5$) | 0.158 | 0.235 | 0.816 |
| 25 | 1397 | 0.048 | 10 ($3d^{10}$) | 0.0284 | 0.0421 | 3.74 |
| 735 MeV $^{208}$Pb ($V_p/V_o$ = 11.89, $E_{KE}$ = 1922 eV) | | | | | | |
| 39 | 1884 | 0.102 | 7 ($4d^7$) | 0.0213 | 0.0316 | 0.693 |
| 47 | 2605 | 0.05 | 17 ($3d^{10}4s^24p^5$) | 0.020 | 0.0299 | 1.05 |
| 50.5 | 3000 | 0.0346 | 13.5 ($3d^{10}4s^24p^{1.5}$) | 0.015 | 0.0227 | 1.23 |
| 55 | 5555 | 0.0042 | 9 ($3d^9$) | 0.0056 | 0.0083 | 1.48 |
| 936 MeV $^{208}$Pb ($V_p/V_o$ = 13.42, $E_{KE}$ = 2448 eV) | | | | | | |
| 23 | 858 | 0.28 | 5 ($4f^5$) | 0.0464 | 0.0687 | 0.114 |
| 40 | 1945 | 0.10 | 6 ($4d^6$) | 0.0285 | 0.0422 | 0.316 |
| 54 | 5414 | 0.0081 | 10 ($3d^{10}$) | 0.020 | 0.0294 | 0.401 |
| 56 | 5703 | 0.0068 | 8 ($3d^8$) | 0.00557 | 0.00824 | 0.855 |

**Table 9.** A summary of LiF sputtering. Ion, energy (E in MeV), incident angle ($\theta_1°$) measured from the surface normal, incident charge ($Q_o$), equilibration length ($L_{ps}$, reaching 90% of the electronic stopping power at the equilibration–charge) [36], ratio (RY) of the sputtering yield under the non-equilibrium charge ($Q_o$) incidence over that the equilibrium-charge ($Q_{eq}$) incidence for isotropic, anisotropic components and total. Lp is the length such that the calculated RY equals to the experimental RY.

| Ion | E (MeV) | $\theta_1$ | $Q_o$ | Lps (nm) | RY$_{ISO}$, Lp (nm) | RY$_{ANI}$, Lp (nm) | RY$_{TO}$, Lp (nm) |
|---|---|---|---|---|---|---|---|
| Au | 200 | 20 | 15 | 18 | 0.0815, 4.3 | 0.923,(176) | 0.226, 11.2 |
| Au | 200 | 60 | 15 | | 0.44, 21.4 | 1.0, | 0.608, 33.2 |
| I | 150 | 70 | 12 | 19 | 0.444, 18.9 | 1.06, | 0.631, 31.2 |
| Pb | 735 | 60 | 39 | 170 | 0.41, 73.1 | 0.64, 174 | 0.507, 109 |
| Pb | 735 | 60 | 47 | 180 | 0.695, 41.3 | 1.17, | 0.894, 290 |
| Pb | 735 | 60 | 55 | 200 | 1.715, 8.0 | 1.666, 16.4 | 1.69, 12.2 |
| Pb | 936 | 60 | 23 | 300 | 0.0181, 21 | 0.0312, 37.2 | 0.024, 29 |

*3.4. Electronic Sputtering of SiO$_2$, UO$_2$ and UF$_4$*

In this section, the current status is briefly described for the charge-state effects of the electronic sputtering of SiO$_2$, UO$_2$ and UF$_4$. In the case of SiO$_2$ sputtering by 50 MeV $^{63}$Cu$^8$ ions (Q$_{eq}$ = 16) (Arnoldbik et al.) [32], the relative sputtering yields RY vary from 0.322 ($\theta_1$ =65°) to 0.858 ($\theta_1$ =85°), $\theta_1$ being the incident angle measured from surface normal and RY follows over cosine, implying that atoms escape easier in the near-surface region as in the case of WO$_3$ (Section 3.2) and LiF (Section 3.3). The single-electron loss cross-section ($\sigma_{L1T}$) is obtained to be 0.6 × 10$^{-16}$ cm$^2$ [63,64] with IP = 199 eV [65] and the characteristic length L is 7.6 nm (2.2 × 10$^{22}$ Si cm$^{-3}$ in SiO$_2$). Hence, the path length contributing to the sputtering is anticipated to be ~7.6 × 2.3 = 17 nm. Arnoldbik et al. also measured sputtered O atoms as a function of Cu$^8$ fluence for various thicknesses of SiO$_2$ films (2–11 nm) at $\theta_1$ =83°. The results imply that the sputtering yield reaches a maximum for the film thickness of 2.5 nm at high ion fluence, where the overlapping effect is not negligible. Nevertheless, the path length is 20 nm corresponding to the thickness of 2.5 nm and this is comparable with the estimated path length of 17 nm mentioned above. Sugden et al. [30] obtained RY of 0.27 by 30 MeV $^{35}$Cl$^6$ (Q$_{eq}$ = 11) at $\theta_1$ = 70°. For 30 MeV $^{35}$Cl$^6$, $\sigma_{L1T}$ and L are obtained to be 0.68 × 10$^{-16}$ cm$^2$ with IP = 114 eV [65] and 6.6 nm, L being consistent with the above result. One notes that sputtering of SiO$_2$ is stoichiometric and the sputtering yields follow S$_e$$^3$ for the equilibrium-charge incidence [31,34].

For UO$_2$, Meins et al. reported the charge- or S$_e$-dependent sputtering yields of U by 5 to 30 MeV $^{35}$Cl ions with the charge of 3 to 6 [18]. Their results seem peculiar that the sputtering yields decrease with increasing S$_e$, contrary to anticipation. In order to analyze their data, sputtering yields of UO$_2$ by ions with the equilibrium charge are required. A large scatter of sputtering yields of U by ions with the equilibrium-charge has been reported (Bouffard et al.) [16] and (Schlutig) [17], e.g., 4.8 × 10$^3$ at S$_e$ = 57.5 keV/nm [16] and 98 at S$_e$ = 55 keV/nm [17]. Hence, the exponent of n for the power-law fit (sputtering yields ~ S$_e$$^n$) varies from 3.5–1.9 and n = 1.9 is derived from [75]. This has to be resolved for further studies. The sputtering yields of U atoms ejected per fission fragment vary 10$^3$ to 110 (Rogers) [13,14] (the variation could be partly explained by grain growth and the lowest yield of 4.5 is more likely due to overlapping effect) and 7 by Nilsson [15]. It would be also interesting to incorporate the results after correcting for the geometrical complexity to usual ion impact sputtering geometry. In addition, it is unknown whether the sputtering of UO$_2$ is stoichiometric or not.

Charge-dependent sputtering yields of UF$_4$ have been reported for $^{19}$F of 0.25 to 1.5 MeV/u (Q$_o$ = 2–9) (Meins et al.) [18]. The exponent of n for the power-law fit (sputtering yields~S$_e$$^n$) under the equilibrium-charge incidence is obtained to be ~3.8 ($^{16}$O, $^{19}$F and $^{35}$Cl of 0.125 to 1.5 MeV/u [18], ignoring the peculiar behavior that the sputtering yields decrease with increasing S$_e$, above 0.5 MeV/u), and n = 4.3 is derived from [75]. Yield by 197 MeV Au [28] would fit the data [18] mentioned above, if a much weaker incident angle dependence of (cos$\theta_1$)$^{0.83}$ is assumed, contrary to the (cos$\theta_1$)$^{1.7}$ (LiF) and (cos$\theta_1$)$^{2.1}$ (NaCl [76]). A large deviation from stoichiometric sputtering has also been reported, F/U~1.7 [28].

*3.5. Surface Morphology Modification: Poly-Methyl-Methacrylate (PMMA), Mica and Tetrahedral Amorphous Carbon (ta-C)*

Charge state effects on the crater formation on PMMA by 593 MeV Au ions with Q$_o$ = 30 to 51 have been observed by Papaleo et al. [10]. S$_e$~Q$_o$$^{1.67}$ has been calculated by CasP 3.2 [10], while CasP 5.2 calculation gives a little bit weaker Q$_o$ dependence, i.e., S$_e$(Q$_o$ = 30) = 11 keV/nm to S$_e$(Q$_o$ = 51) = 22 keV/nm and the exponent is obtained to be ~1.2. Here, the composition and density of PMMA are taken to be C$_5$H$_8$O$_2$ and 1.2 g cm$^{-3}$ (0.72 × 10$^{22}$ mol cm$^{-3}$). The equilibrium charge of 49.1 (C$_5$H$_8$O$_2$) and 48.3 (C$_5$O$_2$) by CasP 5.2 is a little bit larger than that of 46.3 [10]. For grazing incidence ($\theta_1$ =79°), both the crater diameter (~60 nm) and depth (~7 nm) are nearly independent of both Q$_o$ (or S$_e$) and film thickness, indicating that the charge-evolution effect is completed in the path

length of ~60 nm corresponding to the crater diameter, though which is much smaller than 300 nm for 90% equilibration [10]. For normal incidence, however, the situation is different. The crater depth is independent of film thickness and depends on $Q_o$ such that ~2 nm at $Q_o$ = 30 (smaller than that at the grazing incidence) to ~7 nm at $Q_o$ = 51 (comparable with that at the grazing incidence). The crater diameter depends weakly on $Q_o$ (12 nm at $Q_o$ = 30 to 17 nm at $Q_o$ = 51) and film thickness. The diameter is much smaller than ~60 nm at the grazing incidence. These imply the significant charge-evolution effect for normal incidence. Moreover, the effect of the electronic energy deposition on the crater generation is more effective for the grazing incidence (very near-surface effect) and this is to be investigated. The single electron loss cross-section is obtained to be $11 \times 10^{-16}$ cm$^2$ using the empirical formula [63,64] at $Q_o$ = 30 with IP = 868 eV [67] and $N_{eff}$ = 3, and the characteristic length L of the charge-evolution corresponding to the cross-section is calculated to be 1 nm, much smaller than the value of ~130 nm (300 nm/2.3) [10], taking the factor of 2.3 into account as described in Sections 3.2 and 3.3. The appreciable contribution of H is recognized and when the H contribution is discarded, the single electron loss cross-section and L are obtained to be $0.21 \times 10^{-16}$ cm$^2$ and L = 66 nm, L being comparable with the value mentioned above, though no detail description of L estimation [10]. It would be interesting to measure the dependence of crater size on $S_e$ at the equilibrium-charge incidence to compare the charge state dependence. Additionally, measurements of the sputtering yields would be interested to compare with atoms in a crater volume $V_{crater}$. For example, $V_{crater}$ is estimated to be ~$1.3 \times 10^{-18}$ cm$^3$ (crater diameter of d~16 nm and crater depth of z~5 nm at the equilibrium charge $Q_{eq}$ = 46.3) and the number of carbon atoms in the volume (sputtering yields) is calculated to be $4.6 \times 10^4$, employing $3.6 \times 10^{22}$ carbon cm$^{-3}$, if single ion generates one crater as mentioned [10]. $V_{ridge} - V_{rough}$ reads ~$0.5 \times 10^{-18}$ cm$^3$ at the equilibrium charge, $V_{ridge}$ being the ridge volume. A part of atoms in the crater may move to the ridge region.

Alencar et al. have observed the charge-state effect on the formation of hillock or track (crater and ridge) in muscovite mica ($KAl_3Si_3O_{12}H_2$) by 593 MeV $^{197}$Au with $Q_o$ = 30 to 51 [12]. The average hillock diameter of ~25 nm at $Q_o$ = 45 (near the equilibrium charge of 46.3) is similar to that in PMMA mentioned above. The height, diameter and volume of hillock scale $Q_o^{1.5}$ (~1 to 2 nm), $Q_o^{0.53}$ and $Q_o^{3.3}$ (~60 to 450 nm$^3$), respectively. $Q_o$ dependence of the electronic stopping power is $Q_o^{1.2}$ and thus the hillock volume depends on $S_e^{2.2}$. Total atoms in the volume of 240 nm$^3$ ($Q_o$ = 45) is ~$2 \times 10^4$, implying large sputtering yields, even though a considerable fraction remains in the ridge. Again, it would be interesting to measure the dependence of hillock size (height, diameter and volume) as well as sputtering yields on $S_e$ at the equilibrium-charge incidence to compare the charge-state dependence. It remains in question whether the original composition is kept in the ridge induced by ion impact.

In tetrahedral amorphous carbon (ta-C) with SP$^3$ bond fraction of ~80%, conversion into SP$^2$ bond (graphitization) or electrically conducting track formation and hillock formation (its height of several nm) have been reported by Gupta et al. [11] for irradiation at normal incidence by 1 GeV $^{238}$U ($Q_o$ = 26 to 63, correspondingly $S_e$ = 20.5 to 56.6 keV/nm), 997 MeV $^{209}$Bi ($Q_o$ = 26 to 60, correspondingly $S_e$ = 18.9 to 48.5 keV/nm), 950 MeV $^{208}$Pb ($Q_o$ = 23 to 60, correspondingly $S_e$ = 17.2 to 49.6 keV/nm) and 950 MeV $^{197}$Au ($Q_o$ = 26 to 60, correspondingly $S_e$ = 18.2 to 47.9 keV/nm). For Au ions, the hillock height reaches a saturation of ~3.5 nm at $Q_o$ = 52. On the other hand, variation of the track conductivity is quite large and it is difficult to draw systematic dependence on $Q_o$ or $S_e$. One reads from the results [11] that the hillock formation and track conduction becomes appreciable at $Q_o$= ~52 for 950 MeV $^{197}$Au and $Q_o$= ~57 for 977 MeV $^{209}$Bi (Bi data for $Q_o$ = 53 to 56 in the $Q_o$ dependence are not available (Figure 8 [11]) in spite of the AFM image at $Q_o$ = 54 (Figure 4 [11])). The equilibrium charge is estimated to be 52.7 (Equation (2b)) and 54 [58] for 950 MeV Au, and 54.7 (Equation (2b)) and 56 [58] for 977 MeV Bi (54 in [11]). Thus, $Q_o$ =52 (950 MeV $^{197}$Au) and $Q_o$ = 57 (977 MeV $^{209}$Bi) mentioned above are close to the equilibrium charge. Therefore, it is anticipated that the charge-evolution from these $Q_o$ does not play a role as discussed below. IP (ionization potential) and $N_{eff}$ (number of

removable electrons) is obtained to be 5013 eV and 9 for $Au^{52}$ ions [67]. The single electron-loss cross-section [63,64] for 950 MeV $Au^{52}$ ions is obtained to be $0.724 \times 10^{-16}$ $cm^2$. The cross-section of multi-electron loss [62] is larger by ~45% than that of single-electron loss. Hence, the characteristic length for the charge-evolution is less than 0.9 nm (carbon density of $1.5 \times 10^{23}$ $cm^{-3}$, 3 $g\ cm^{-3}$ [11]) and the characteristic length seems to be small so that the charge-evolution is ineffective. For 977 MeV $Bi^{57}$ ions, the single electron–electron capture cross-section is obtained to be $0.418 \times 10^{-16}$ $cm^2$ [68] and correspondingly the characteristic length is 1.6 nm. Hence, the same argument of 950 MeV $Au^{52}$ ions holds for Bi ions.

## 4. Discussion

Relative yields (RY) of non-equilibrium-charge incidence over those of the equilibrium-charge incidence are calculated with the experimentally observed dependence of the electronic-excitation effects on the electronic stopping-power ($S_e$) at the equilibrium charge, empirical charge-changing cross-sections and theoretical charge dependence of $S_e$ (CasP code). In Section 3.1, Section 3.2, Section 3.3, charge state effects are described for lattice disordering of $WO_3$ and electronic sputtering of $WO_3$ and LiF. It is shown that the simple model is able to explain the experimental results to some extent, however, the explanation may not be adequate especially for the electronic sputtering. The speculation is that charge-changing processes may have roles in the charge-state effects in addition to the mechanism via electronic stopping power $S_e$. Experimentally, thickness dependence of the electronic sputtering and electronic-excitation induced material modifications such as lattice disordering is strongly desired under non-equilibrium charge incidence as well as equilibrium-charge incidence.

Charge state effects on the electronic sputtering of $SiO_2$, $UO_2$ and $UF_4$ are briefly mentioned in Section 3.4. At grazing incidence, sputtering yields of solids mentioned above, as well as LiF, are enhanced and this "near-surface effect" is to be studied. Additionally described is the charge state effect on surface morphology (topology) modification of PMMA and mica, and bond modification (transformation of $sp^3$ bond into $sp^2$ bond) of ta-C resulting in an increase in the electrical conductivity. Comparison is desired with those of the equilibrium-charge incidence.

One of the problems for studying the charge-state effects is the accuracy of empirical formulas for the charge-changing cross-sections in solids, since these are known to limited solids, e.g., carbon. Solid-state effects on charge-changing cross-sections are also to be investigated, though the effects are anticipated to be small, because inner shells are involved in the charge-changing processes for highly charged ions and basically, they have no phase effect. There are three models for atomic displacement induced by the electronic energy deposition: Coulomb explosion [5,6], Thermal spike [34,35,75] and Exciton [23,77,78]. A mechanism has been discussed for electron-lattice coupling that displacement of lattice vibration amplitude (~ one-tenth of the neighboring atomic separation in solids) from the equilibrium position can be achieved by Coulomb repulsion during a short neutralization time of ~10 fs in the positively charged region generated by high energy ion impact. Eventually, this results in highly-excited states coupled with lattice in the electronic system, h-ESCL (equivalent to multi-exciton coupled with lattice), and non-radiative decay of h-ESCL leads to atomic displacement (exciton model [37,44,48]. Further studies of the charge-state effects would help in the understanding of the mechanism of electronic excitation effects which has not yet been established.

## 5. Conclusions

Charge state effects on material modifications induced by electronic excitation such as electronic sputtering, lattice disordering and surface morphology modification have been described. At the equilibrium-charge incidence, lattice disordering of $WO_3$ and the sputtering yields for a variety of non-metallic solids scale with the electronic stopping power ($S_e$) and the information is important for studying the charge-state effects. It

is shown that the saturation approximation to the charge-evolution with the empirical formulas of charge changing cross-sections and charge-dependent $S_e$ reasonably explains the lattice disordering of $WO_3$. However, the explanation with the simple model is not adequate for the charge-state effect on the electronic sputtering of $WO_3$ and LiF. Thickness dependence of the charge-state effects would give an insight into solid-state effects on the charge-changing process and mechanism of the effects induced by electronic excitation or the energy transfer from the electronic system into the lattice.

**Author Contributions:** Conceptualization and writing, N.M.; investigation, M.S., S.O. and B.T. All authors have read and agreed to the published version of the manuscript.

**Funding:** This research received no external funding.

**Data Availability Statement:** All data in this manuscript are from the published papers.

**Conflicts of Interest:** The authors have no conflict.

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
