# Peer review of "Charge State Effect of High Energy Ions on Material Modification in the Electronic Stopping Region"

_atoms, doi:10.3390/atoms9030036_

Round 1

Reviewer 1 Report

The article is scholarly and I recognize the extensive references to relevant literature on charge-state effects on particle-material interaction. I am convinced that the article merits publication in MDPI Atoms.

That said, the actual model that is developed in the article remains obscure to me. I urge the authors to include within section 2 a clear and coherent statement, in one well-defined location, of the model. I will explain my confusion.

There are equations 1a and 1b as alternatives for Q_{eq} and equations 2a and 2b as alternatives for Q_m, and within 2a or 2b there are free parameters L or (L1 and L2). There is a power-law expression for S_e with a free coefficient k in the exponent and a free amplitude in front. There is a free exponent n in equation 3a for RY.

So what is finally the model? Which of the equations 1a and 1b, which of 2a and 2b, what values of the free parameters. I don't believe that one can treat these parameters as free for each of the five applications separately; then it is just parameter fitting one application at a time, not a model. The requested statement of the model needs to be in one place, say at the end of section 2, and not distributed over the entire section.

With that issue addressed the manuscript is, in my opinion, suitable for publication in Atoms.

Reviewer 2 Report

Referee report on review paper “Charge State Effect on Material Modification” by Noriaki Matsunami, Masao Sataka, Satoru Okayasu, Bun Tsuchiya

The paper is a review of recent results related to studies of the influence of the projectile charge state and its evolution during the collisions of swift heavy highly charged ions with non-metallic solids on the resulting material modification. The authors show that a simple analytic model with 1-4 fitting parameters may provide a satisfactory description of the experimental data on X-ray diffraction intensity degradation, as well as on some of the sputtering induced by ion impact. The role of the charge state of the projectiles is demonstrated. The research area develops rapidly in past two decades and, to referee’s knowledge, no recent review papers devoted to the charge state effects specifically were published. Thus, a review paper on this subject is likely to be appreciated by the readers. The authors of the paper are known specialists in the respective field of research, have many recent works published and are undoubtedly able to provide a comprehensive review of the recent advances in the selected area.

The review presents a number of interesting and important results and covers a great deal of works published by authors themselves as well as by other researchers. At the same time, the style of the presentation of the results is, in referee’s opinion, more suitable for a handbook chapter rather than for a review paper. Namely, the authors mostly “list” the results and focus on how the results of fitting and calculations agree with the experimental data, while keeping the physical processes and mechanisms governing the phenomena out of discussion. Review papers should provide a broader perspective of the field and should have at least some discussions of the fundamental concepts so that even a reader currently not working in the field can become familiar with most important advances in the respective area.

Listed below are comments which support this position:

  1. Right in the Introduction the authors start discussing charge state effects, but do not give a clear definition of what they call charge state effects. Some illustrative examples of how the ion charge changes to equilibrium charge could be given here. Moreover, even the terms of “equilibrium charge” (and, specifically, that it often depends on the material), “stopping power” (including its possible angular dependence) and sputtering yields are not explained well. This may be confusing for the potential readers not familiar with the terminology adopted in the field.
  2. The elementary processes which lead to the equilibrium charge formation are only mentioned in the text. It would make sense to explicitly list the processes preferably with some data/refs on the respective cross sections.
  3. The paper features no discussion of the experimental methods / setups used to obtain the experimental data presented. Adding the corresponding section would greatly contribute to making the paper more “readable”.
  4. The theoretical model used in a central point of the review. Currently its description is minimalistic. Some discussions of the foundations of the model, approximations used, estimated accuracy and possible enhancements are needed.
  5. The computational codes used are barely discussed at all. It is preferable to give at least a brief discussion of the approaches realized in the codes.
  6. In Section 3 the authors mainly list the results obtained and primarily focus on the agreement between the theory and the experiments and how that depends on the fitting parameters. Such information is very important for specialists working in the field. However, a general reader specializing in atomic physics would be much more interested in the discussion of the physical processes and phenomena which result in the experimental and theoretical dependences presented. With a couple of exceptions, such discussion is not given. In particular this refers to lines 270-291, 335-348, 405-409, 501-523, 526-559.

There are also some specific comments which need corrections/improvements in the text:

  1. Eqs. (1a)-(1b) are approximate, and contain adjustable parameters. In such situation it makes sense to either provide an acceptable range of values of the adjustable parameters, or to give a corresponding confidence interval.
  2. In Eqs. (1a)-(1b) the quantity Vp is not defined.
  3. In Section 2 the authors make use of saturation approximation, but neither discuss it, nor give references to works where the approximation is explained.
  4. Throughout the text the positive ions are denoted by A+q while in standard notation they are denoted by Aq+.
  5. When dealing with 2-exponent-based model, and, in particular, when obtaining the magnitudes of the lengths in the exponents, virtually no discussion of how arbitrary the lengths in exponents are is provided. Since the lengths are most commonly obtained from the cross sections, and the latter are often not known with sufficient precision, the resulting lengths may be subject to some variation.
  6. Term “relative yield” on page 4 lacks clear definition.
  7. When discussing X-ray diffraction intensity degradation, the authors compare the magnitude of the intensity corresponding to a given angle. Since, as indicated by Fig. 2a, the diffraction has some angular dependence, one needs to clarify how the degradation is calculated: from the ratio of the magnitudes of the intensity in the peak position, or from the ratio of the integrals over the angle in the vicinity of the peak.
  8. On lines 379-428 the authors discuss anisotropic sputtering. As anisotropy can be quite different, it would be preferable to provide some illustrative information on the angular dependences obtained.
  9. On line 558 the quantity of Vridge is not defined.

While in general the paper is well referenced, adding references to the following well-known and recent works in the field is needed:

https://www.mdpi.com/2218-2004/9/1/17/htm

https://journals.aps.org/pra/abstract/10.1103/PhysRevA.101.012704 (some discussion of BREIT code could also be added)

https://doi.org/10.3367/UFNe.2016.10.038012

https://doi.org/10.1098/rsta.2003.1300

https://doi.org/10.1103/PhysRevLett.107.063202

https://www.springer.com/gp/book/9783540445005

https://doi.org/10.1016/j.nimb.2018.04.002

https://doi.org/10.1016/j.carbon.2015.12.101

https://doi.org/10.1103/PhysRevB.73.184107

https://doi.org/10.1016/j.surfrep.2010.11.001

The paper needs a spelling check (for example, in first paragraph, “On the other hand, the inelastic or electronic collisions (excitation of electrons and ionization) generally end up de-excitation (with or without radiation emission) and heating of materials. However, for the ion with the energy is larger than ~ 0.1 MeV/u, …”. The referee leaves this to Editors discretion.

Recommendation: While the authors indeed reviewed an extensive number of recent results. The authors should amend the manuscript in accordance with the comments above. Particular effort should be put to highlighting the physical processes and mechanisms responsible for the charge state effects discussed. Physical meaning of the respective changes in the experimental and fitting curves presented should be explained in more detail.

Round 2

Reviewer 2 Report

Referee report on the revised version of a review paper “Charge State Effect of High Energy Ions on Material Modification in the Electronic Stopping Region” by Noriaki Matsunami, Masao Sataka, Satoru Okayasu and Bun Tsuchiya

In the revision the authors significantly improved the quality of the manuscript making it much easier to understand to the readers of Atoms. Most of the comments and concerns of the initial review were addressed. However, I believe that 3 points of the initial review still need additional clarifications in the text:

  • Original review: “The elementary processes which lead to the equilibrium charge formation are only mentioned in the text. It would make sense to explicitly list the processes preferably with some data/refs on the respective cross sections.”. Authors’ response: “The equilibrium-charge is determined by solving the rate equation using the charge changing (electron loss and capture) cross sections and the present paper does not concern the details of the processes (e.g., electron loss is the ionization of projectile ion due to interaction with target atoms in the material, quite similar to electronic stopping of projectile ion, where excitation and ionization of the target atoms are involved).”

Comment: I agree that such approach can be used with the model proposed by the authors. At the same time, I believe that the authors should explicitly state this in the manuscript as in atomic physics the terms of “electron capture” and “electron loss” are often used to describe specific physical processes, while the authors use these terms to describe rather broad ranges of processes leading to ion charge decrease and increase. This may result in some confusion.

  • Original review: “Throughout the text the positive ions are denoted by A+q while in standard notation they are denoted by Aq+.”. Authors’ response: “In this paper, positive ions are treated. Aq+ may be standard in the atomic physics. However, we find, for example, that S+1 is easier to read than S1+ and A+q is employed in the present paper.”

Comment: As far as I understand, the authors prefer “+q” notation over “q+” because of the charge fractions which indeed look better like “Qm=+7” vs “Qm=7+”. However, the authors only consider positive ions (cations) so that one can just write “Qm=7”. Thus, I believe that in accordance with aims and scope of Atoms journal, it is better to change the manuscript to standard notation adopted in atomic physics.

  • Original review: “The paper features no discussion of the experimental methods / setups used to obtain the experimental data presented. Adding the corresponding section would greatly contribute to making the paper more “readable”.”. Authors’ response: “The present paper deals with the charge-state effects of ions in the electronic stopping region, based on the published experimental results and avoids the results, if fatal problem in the experimental setup or large deviation from the anticipation (or published results) is found. Thus description for only basic or general experimental setup is given. Minimum information of materials has been added as follows….”

Comment: I understand the authors’ point. Still, I believe that at least 1 paragraph citing different experimental approaches utilized in the field needs to be included in the introduction, probably after line 102.

Recommendation: The authors greatly improved the manuscript. I believe it can be accepted for publication in Atoms after the authors make minor changes in their paper in accordance with the comments above.

Author Response

Thank you for the comments again. Revisions are as follows (in blue below and red in the revised manuscript).

  1. Increase and decrease of the ion charge are naturally the result of the electron loss and capture processes. In this paper, we do not concern the specific states related to these charge changing process, but only the mean-charge evolution of ions, in spite that the many states (e.g., atomic levels in the limited use) are involved. Experimentally, incident-charge of ions is varied to study the charge state effect. Use of the mean-charge intends that the details of the processes in the specific state are discarded. At present, there seem no implications of the effects due to a specific state.
  2. Both “+7” and “7+” have the same meaning. We prefer the former because of easier reading, especially in the small letters of superscript. (Qm, Qo and Qeq>0) has been added in the 1st paragraph, section 2 and we have removed “+” as possible as we could in the text.

3. Very brief descriptions of the experimental for surface morphology observations and sputtering measurements have been added in the 1st paragraph, Introduction